# Attentional Engagement and Disengagement Differences for Circumscribed Interest Objects in Young Chinese Children with Autism

**DOI:** 10.3390/brainsci12111461

**Published:** 2022-10-28

**Authors:** Li Zhou, Li Zhang, Yuening Xu, Fuyi Yang, Valerie Benson

**Affiliations:** 1Faculty of Education, East China Normal University, Shanghai 200062, China; 2Jiangsu Key Laboratory for Big Data of Psychological and Cognitive Science, Yancheng Normal University, Yancheng 224000, China; 3Faculty of Psychology, Tianjin Normal University, Tianjin 300387, China; 4School of Psychology and Computer Science, University of Central Lancashire, Preston PR1 2HE, UK

**Keywords:** autism spectrum condition, attentional engagement and disengagement, circumscribed interest, gap-overlap paradigm, saccade, eye movement, modified gap-overlap paradigm

## Abstract

The current study aimed to investigate attentional processing differences for circumscribed interest (CI) and non-CI objects in young Chinese children with autism spectrum condition (ASC) and typically developing (TD) controls. In Experiment 1, a visual preference task explored attentional allocation to cartoon CI and non-CI materials between the two groups. We found that ASC children (*n* = 22, 4.95 ± 0.59 years) exhibited a preference for CI-related objects compared to non-CI objects, and this effect was absent in the TD children (*n* = 22, 5.14 ± 0.44 years). Experiment 2 utilized the traditional gap-overlap paradigm (GOP) to investigate attentional disengagement from CI or non-CI items in both groups (ASC: *n* = 20, 5.92 ± 1.13 years; TD: *n* = 25, 5.77 ± 0.77 years). There were no group or stimulus interactions in this study. Experiment 3 adopted a modified GOP (MGOP) to further explore disengagement in the two groups (ASC: *n* = 20, 5.54 ± 0.95 years; TD: *n* = 24, 5.75 ± 0.52 years), and the results suggested that exogenous disengagement performance was preserved in the ASC group, but the children with ASC exhibited increased endogenous attentional disengagement compared to TD peers. Moreover, endogenous disengagement was influenced further in the presence of CI-related objects in the ASC children. The current results have implications for understanding how the nature of engagement and disengagement processes can contribute to differences in the development of core cognitive skills in young children with ASC.

## 1. Introduction

Autism spectrum condition (ASC) is a lifelong neurodevelopmental condition principally characterized by impairments in social interaction and communication and the presence of restricted interests and repetitive behaviors [1]. Though differences in the deployment of attention in ASC are not formally diagnostic features, such differences are still commonly observed. For example, there is evidence for diminished attention to the face (especially the eyes) and evidence to support enhanced attention to some non-social domains [2,3,4,5]. Reports also show that early attentional impairments might exert a profound influence on children’s later cognitive processing development [6,7], and this includes but may not be limited to the domains of social engagement, joint attention, emotional expression and processing, and language acquisition [8,9,10]. Indeed, strong evidence confirms that infants with ASC exhibit early attentional impairments, and such processing differences could be used effectively as one of the earliest biomarkers of ASC [8,11,12].

### 1.1. Attentional Disengagement and the Gap-Overlap Paradigm (GOP)

It is well established that spatial orienting of attention comprises three distinct operations: disengagement from the current focus, shifting attention to a secondary locus, and engagement of attention with the new target/location [13]. Effective disengagement serves as the basis of the subsequent two attentional processes and thus is essential for individuals to acquire new information seamlessly. A growing number of oculomotor studies have reported that individuals with ASC struggle to disengage from fixated stimuli when they make overt eye movements to new targets [7,14,15,16,17]. However, other studies (e.g., [18,19,20,21,22]) report no evidence for disengagement difficulties in ASC. Furthermore, convergent findings show that in the first year of life, through the identification of longer saccadic reaction times (SRT), high-risk autistic infants had difficulty disengaging from currently fixated items compared to low-risk infants [8,23]. Therefore, the attentional disengagement hypothesis (dysfunctional attentional disengagement may be one of many primary characteristics associated with the emergence of core ASC symptoms) has important implications for clinical purposes (diagnostics) and autistic cognitive theory [6].

The gap-overlap paradigm (GOP) has been adopted to investigate attentional disengagement ability in individuals with ASC [24,25]. In this paradigm, three conditions (baseline, gap, and overlap) have been adopted to assess disengagement ability [24,25,26]. In the baseline and gap conditions, the central stimulus is automatically disengaged because the stimulus is extinguished before the target appears, which reveals participants’ exogenous attentional disengagement abilities. In the overlap condition, the central fixation stimuli remain on the screen and must be intentionally disengaged to orient to a new target presented in the periphery. This results in longer SRT and reveals participants’ endogenous attentional ability [21,25,27]. The SRT, disengagement cost, and failure-disengagement rates for the GOP paradigm are critical measures to highlight participants’ disengagement characteristics [24,28]. Additionally, the disappearance of the central stimulus in the gap (and, to some extent, the baseline) condition has been shown to function as a spatially non-specific warning cue that decreases participants’ reaction speed and leads to a transient increase in arousal. Therefore, more pronounced SRT differences between overlap and gap (baseline) trials could also reflect impaired arousal regulation and alterations in the alerting network [6,7,24,29].

### 1.2. Attentional Disengagement and Potential Influencing Factors in Autistic Studies

Landry and Bryson [15] initially demonstrated difficulties with attentional disengagement in young children with ASC using the GOP task. They found that the ASC children took much longer (as evidenced by delayed SRT) to disengage from central objects compared to typically developing (TD) children in the overlap condition (where the central stimuli and the lateral target were presented at the same time). Low-functioning ASC adults have shown disengagement difficulties, as suggested by electrophysiological measures in the GOP task [27]. Intuitively, this disengagement difficulty should continue with development in ASC, as previous studies have reported it as a reliable early precursor. However, some evidence does not support this conclusion (e.g., [18,19,20,21,22]). For example, van der Geest et al. [30] found that children with ASC performed similarly on an attentional disengagement task relative to their TD peers. Fischer et al. [19,20] also found intact attentional disengagement performance in toddlers and children with ASC. ASC children with mixed abilities, such as moderate or severe verbal communication impairments, have also shown preserved visual disengagement [22]. These intact disengagement reports are inconsistent with the early atypical performance related to visual disengagement in the first year of life. Two potential factors discussed below, which could lead to conflicting reports on attentional disengagement in ASC, include the adoption of the classic GOP task and the stimulus types employed in previous studies.

Using the classic GOP task, some prior studies reported no group differences in attentional disengagement (e.g., [18,30,31]). However, in the overlap condition of the traditional GOP, the foveal stimulus is presented first on its own. Then it overlaps in time with the presentation of the peripheral target in the following display in a single trial [18]. In this condition, regardless of the participant group, there is adequate time to fully process the foveal stimuli so that attention and eye movements can quickly be initiated to the target when the target appears. Thus, both ASC and TD individuals may be able to disengage their attention and show similar SRT, disengagement cost, and failure-disengagement rates under the classic GOP condition (e.g., [18,19,20,30]). 

Several previous studies [19,20,27] utilized many different kinds of non-social images as centrally fixated stimuli, such as vehicles, fruits, or baggage, without controlling for whether any of the stimuli might be of particular interest to ASC children. A lack of control over the type of stimuli presented could also be a factor that has contributed to the mixed findings reported in studies that investigate attentional disengagement in ASC, as it is well documented that individuals with ASC have circumscribed interests (CI). CI is included in the diagnostic criteria listed within the restricted and repetitive behaviors domain, and commonly occurs in individuals with ASC. CI is defined as a functionally impaired preoccupation with a narrow range of topics [32,33]. In an initial study, South et al. [34] reported that eight common categories of CI objects, manifest in individuals with ASC, including an extreme interest in trains, vehicles, planes, electronics, blocks, computer equipment, road signs, and sporting equipment. Non-CI items less likely compelling to ASC individuals included clothing, outerwear, office supplies, kitchen supplies, furniture, tools, musical instruments, and plants. Children with ASC exhibited a more perseverative, focused, and circumscribed attentional style for CI-specific objects [35,36]. Increased visual attention to CI-related objects has also been shown in young children with ASC [5,37]. Specifically, more attention was devoted to the CI items when scanning simple or complex image arrays compared to the non-CI items. However, TD children did not differ in the amount of attention devoted to these two stimuli types, presenting a chance-level looking pattern between CI and non-CI objects. Furthermore, in an anti-saccade task [38], in which the goal is to make an eye movement to the opposite location to a target, ASC children and adolescents showed reduced attentional control for CI-related objects (but not for non-CI-related objects), making more erroneous eye movements towards (instead of away from) the CI items compared to their TD peers. The above findings suggest that children with ASC would show greater attentional engagement when they fixate on CI-related stimuli, potentially resulting in delayed disengagement speed to a newly presented target appearing in the periphery. Previous studies have tended to include both CI and non-CI objects together (e.g., [19,20,27]) in the stimulus set, which means that the influence of CI-related (non-social) stimuli remains to be formally investigated in detail concerning attentional engagement and disengagement processes in ASC.

### 1.3. The Present Study

The current study used the same cartoon stimulus set to investigate attentional engagement and disengagement for CI and non-CI objects in young Chinese children with and without ASC. In accordance with the visual preference evidence related to CI stimuli presented earlier, we predicted that stimuli type would modulate attentional engagement and disengagement in ASC children. Based on previous evidence to support delayed attentional disengagement in ASC [8,14,15,27], we also predicted that the ASC group would show differences in disengagement from fixated CI-related items compared to non-CI-related items. The effects observed for the ASC children were expected to be absent in the TD children. 

Three experiments were included in the current study. Experiment 1 aimed to investigate whether young children with ASC have an attentional bias to engage more with CI-related compared to non-CI-related objects, in line with prior reports [5,37,39]. A free viewing preference task was conducted to achieve the aim of Experiment 1. Free viewing ensured that task demands did not influence attentional deployment. Experiment 2 aimed to investigate whether the same stimuli used in Experiment 1 would influence disengagement in children with ASC. The classic GOP paradigm was adopted to achieve the aim of Experiment 2, and we used similar procedures to those employed in studies that have reported disengagement effects using eye movement metrics in ASC [19,20]. Experiment 3 aimed to investigate whether disengagement effects in ASC would be more readily observed using a modified GOP (MGOP) version of the classic paradigm (which has not always provided evidence for disengagement difficulties in ASC) [18,30]. In the MGOP, participants will be prevented from pre-processing the CI and non-CI stimuli prior to target presentation in the overlap condition.

The rationale for using the same stimuli in all studies was to show that an attentional bias to engage with CI-related stimuli (Experiment 1) should also result in difficulties in disengaging (Experiments 2 and 3) when the same CI-related stimuli were presented at fixation. The task was to move the eyes to a newly presented peripheral target. The rationale for using two different disengagement paradigms was to show that one possible reason for the inconsistencies in the literature on disengagement in ASC might reflect differences in the overlap condition in these paradigms. In the classic overlap condition (Experiment 2), the foveal stimulus is presented independently and with the peripheral target in the following display sequence of each trial. This has the potential to mask any group differences in the ability to disengage from the central stimulus since this stimulus can be fully processed before the target display onset. In the modified-overlap condition (Experiment 3), more effort is required to endogenously disengage from the central stimulus because there has been no opportunity to process the foveal stimuli before the target display. Therefore, in contrast with the classic GOP, the opportunity to prepare an intentional saccade before the target’s appearance is absent in the MGOP paradigm. Therefore, we predicted that the novel MGOP would be more effective in revealing actual disengagement differences in eye movement patterns in young children with and without ASC. Furthermore, if attentional engagement biases are observed for CI-related objects in ASC (Experiment 1), we expect stimulus interest to modulate endogenous disengagement.

## 2. Experiment 1: The Visual Preference Task

### 2.1. Methods

#### 2.1.1. Participants

The current experiment was a simple free-viewing task that enabled ASC children of varying levels of function to be included in Experiment 1. The final sample consisted of 22 young Chinese ASC children (two females, M_age_ = 4.95) and 22 age-matched Chinese TD children (two females, M_age_ = 5.14). A power analysis was conducted using G*Power software [40] to determine whether a sufficient number of participants were recruited for Experiment 1. The parameter settings were in line with the published evidence [41], with 0.8 power, an alpha of 0.05, and 0.5 as correlations among repeated measures. At least 34 participants were identified as necessary by a repeated measures ANOVA (RM-ANOVA) with a group (ASC, TD) as the between-group factor and stimulus type (CI, non-CI) as the within-group factor.

The ASC children were recruited from an autism research service center, and the TD counterparts were recruited through a typical preschool institution in Tianjin, China. The ASC children were previously diagnosed by more than one experienced clinician following the diagnostic criteria listed in the DSM-5 [1]. Young children in the TD group reported no history of brain damage and no neurodevelopmental disorders by their parents. The Chinese version of the Autism Spectrum Quotient: Children’s Version (AQ-Child) [42] was adopted to confirm the ASC diagnosis. Children with ASC scored significantly higher in AQ than their TD peers (the cut-off score is 76), *t* = 9.23, *p* < 0.001. Their verbal IQ was tested using the Chinese Version Peabody Picture Vocabulary Test-Revised [43], and there was no significant difference in verbal IQ between groups, |*t*| < 1.96, *p* > 0.05. Written consent was obtained from the participants’ parents before the formal experimental procedure. Before the formal experiment, all children were asked whether they wanted to “look at some cartoon images”. If the child agreed, they would participate in the experiment(s). Detailed information on participant demographics can be found in Table 1. 

#### 2.1.2. Materials and Task

The visual preference task was adopted in Experiment 1. In this task, two pictures were presented, one on the right and one on the left side of the display screen. Materials included 20 static high-quality cartoon picture arrays collected from the Internet, with the same size of 300 × 200 pixels for each. Each array was presented consisting of one CI-related picture and one non-CI picture. The operational definition and selection of both types of pictures have been documented in previous studies [5,34,37,38,39], including CI-related objects such as airplanes, cars, trains, clocks, and non-CI objects such as plants, clothes, and furniture. Stimuli across the two categories appeared on the right and left sides of the display an equal number of times. In total, there were 20 trials in Experiment 1.

#### 2.1.3. Apparatus

A Tobii TX300 eye tracker (Tobii Technology, Stockholm, Sweden) was adopted to record participants’ eye movements, with a sample rate of 300 Hz. A 23-inch TFT monitor (1920 × 1080 pixels, 60 Hz refresh rate) was utilized to display the picture arrays. During the formal test, participants used a chinrest to maintain head stability. The position of the children’s eyes was approximately 65 cm away from the center of the screen.

#### 2.1.4. Procedure

Before the formal test, a five-point calibration procedure was run whereby participants were asked to stare at each of the fixational dots shown at different locations on the screen. This procedure aimed to ensure that participants’ eye positions could be recorded accurately during the formal test. Following calibration, a red crosshair (50 × 50 pixels) was presented at the center of the display for 1000 ms, during which participants were asked to look at it. Next, the paired CI-related and non-CI pictures were displayed synchronously for 5000 ms. The children were instructed to fixate on the central red crosshair and then watch the picture arrays on the screen freely. Finally, a blank screen was presented for 1000 ms. See Figure 1 for a trial sequence example in the visual preference task.

#### 2.1.5. Eye Movement Measures and Data Analysis

Three areas of interest (AOI) were created for each stimulus array using the Tobii Studio software (version 3.2.2; Tobii Technology, Stockholm, Sweden), which were the left-side item AOI (400 × 300 pixels), the right-side item AOI (400 × 300 pixels), and the whole screen AOI (1920 × 1080 pixels). Fixations were calculated based on the Tobii I-VT fixation filter [44] with the following parameter settings: (1) fixations close spatially and temporally (<0.5°, <75 ms) were merged in order to avoid longer fixations from being separated into shorter fixations because of noise or data loss; (2) the velocity threshold was set at 30°/s; (3) duration threshold was set to 60 ms. 

According to previous studies [35,37,39,45], the following eye movement metrics were analyzed in the current experiment to indicate participants’ attentional allocation to both types of stimuli: (a) Preference: The proportion of fixation time devoted to each stimulus type AOI relative to the total time spent on the whole screen, which qualifies the attentional distribution across the two types of objects. (b) Detail Orientation: The fixation counts children made on each stimulus category AOI, revealing the number of explorations in detail. (c) Prioritization: The latency (the time taken from the onset of the display to the first fixation that landed on each stimulus category AOI), measures attention capture and orienting in the early stages of attentional location and processing.

Exclusion Criteria. Trials were excluded when: (a) the entire screen looking time was less than 2000 ms (6.36% for three metrics) [46], (b) eye movement data was missing (3.24% for three metrics), and (c) Preference (0%), Detail orientation (0.11%) and Prioritization (2.16%) were greater or lower than three standard deviations away from the mean value of each participant. There were 1591 valid trials used for the Preference analysis, 1589 usable trials for the Detail Orientation, and 1553 trials for the Prioritization analysis.

Data included in the analyses were analyzed in RStudio (version 1.4.1717), using linear mixed-effects models (LMM) from the lme4 package (version 1.1-27.1) [47]. First, the group (ASC, TD) and stimulus type (CI, non-CI) were defined as fixed factors in the LMM analysis. Then, the random effects for the fixed effects were fitted over participants and trials. Absolute values of the *t*-value equal to or greater than 1.96 indicate a significant difference. 

### 2.2. Results

#### 2.2.1. Preference

No group difference was found for this measure (|*t*| < 1.96, *p* > 0.05). A significant effect of stimulus type was found (*b* = −0.08, *SE* = 0.01, *t* = −8.59, *p* < 0.001), suggesting that participants showed increased fixational distribution for the CI-related items compared with non-CI items. However, a significant group by stimulus type interaction (*b* = 0.13, *SE* = 0.02, *t* = 6.83, *p* < 0.001) indicated that the stimulus type effect on the Preference measure was mainly driven by the ASC individuals, showing greater attentional distribution to the CI-related objects compared to non-CI items (*b* = −0.15, *SE* = 0.02, *t* = −10.09, *p* < 0.001), and this effect was absent in the TD group (|*t*| < 1.96, *p* > 0.05). Figure 2a illustrates the group differences for the Preference measure.

#### 2.2.2. Detail Orientation

A group effect was observed for fixation counts (*b* = 0.91, *SE* = 0.25, *t* = 3.58, *p* < 0.001), showing that children with ASC had fewer fixation counts than their TD peers. The stimulus type also yielded a significant effect (*b* = −1.13, *SE* = 0.14, *t* = −8.34, *p* < 0.001), suggesting that all children made more fixations on CI-related items relative to non-CI items. The group by stimulus type interaction was also significant (*b* = 1.34, *SE* = 0.27, *t* = 4.92, *p* < 0.001), and the simple effects analysis showed that the stimuli type effect was more robust in the ASC group (*b* = −1.80, *SE* = 0.20, *t* = −9.18, *p* < 0.001), relative to the TD group (*b* = −0.47, *SE* = 0.19, *t* = −2.48, *p* = 0.013), although both ASC and TD groups made more fixation counts on CI-related objects than non-CI items. Figure 2b illustrates the group differences for the Detail Orientation measure.

#### 2.2.3. Prioritization

A group effect was significant for this measure (*b* = −0.14, *SE* = 0.04, *t* = −3.44, *p* = 0.001), which showed that the TD group was quicker to orient their first fixation to the AOI stimuli compared to the ASC group. The effect of stimulus type was also significant (*b* = 0.17, *SE* = 0.04, *t* = 4.35, *p* < 0.001), showing that CI-related items caught children’s attention more quickly than non-CI items. However, a group by stimulus type interaction was also significant (*b* = −0.22, *SE* = 0.08, *t* = −2.75, *p* = 0.006), and this indicated that the stimulus type differences over the orientation priority was observed in children with ASC (*b* = 0.28, *SE* = 0.06, *t* = 4.31, *p* < 0.001), but not in the TD group (|*t*| < 1.96, *p* > 0.05). Figure 2c shows the group differences for the Prioritization measure. Table 2 presents the fixed results from LMM analyses in Experiment 1.

### 2.3. Summary

Using the visual preference task, Experiment 1 aimed to investigate attentional engagement differences for CI and non-CI cartoon objects in young Chinese children with and without ASC. Consistent with previous predictions, the current experiment revealed a higher visual preference for CI-related items than non-CI items in the ASC group. This result was supported by all three eye movement measures of Preference (proportional fixation duration), Detail Orientation (total fixation counts), and Prioritization (latency to the first fixation to the AOI item). Moreover, a reduced stimulus type effect was also observed in the TD group, whereby the stimulus type difference was only observed for the measure of fixation counts in the TD group. However, this effect was minor relative to the effect observed in the ASC group. 

The current experiment, which utilized cartoon images, has yielded similar findings to prior studies using actual object pictures as materials. The findings from this and previous studies [5,37] suggest an over-focused attentional engagement to some specific non-social stimuli categories in young children with ASC. For example, Sasson et al. [37] utilized eye-tracking to quantify discrete aspects of visual attention to picture arrays with a sample of 2 to 5-year-old children with and without ASC and found that ASC participants exhibited greater constant attention on CI-related objects than on non-CI objects, which is in line with the current evidence. Further, the CI effect in ASC is unlikely to be driven by the sensory properties of the images, since both prior (e.g., [37,39]) and the current research found a strong CI bias across different stimulus categories (vehicles, plants, clothing, furniture, etc.) and different physical characteristics (colors, shapes, lightness, etc.) of the stimuli.

This atypical visual preference for CI objects can impact cognitive development in young individuals with ASC. The existence of these stimuli, commonly found in daily life, could capture and hold ASC individuals’ attention at the expense of attention devoted to events occurring in the social domain. This attentional bias for CI items could impede the normal detection, orientation, and/or processing of critical social cues [5,38,39] and prevent typical developmental responses to such cues in children with ASC. As noted previously, attentional disengagement is an essential prerequisite for shifting attention from one item and then engaging attention with a new item within one’s environment [13]. Using the same stimuli as in Experiment 1 provides a good test of the validation of the classic GOP task, which has had mixed results. Given the strong attentional biases shown for the CI-related stimuli in Experiment 1, it would be expected that the same stimuli would be more difficult to disengage from in ASC. No other studies to date have used the same stimuli to show attentional engagement and disengagement differences using different paradigms in ASC. Experiment 2, therefore, aimed to explore the influence of CI-related and non-CI stimuli (from Experiment 1) on attentional disengagement processes in both ASC and TD children. According to the current findings and previous reports [5,7,15], we hypothesized that CI-related items would disrupt the disengagement process in young ASC children compared to non-CI items, and that this effect would be absent in the TD group. 

## 3. Experiment 2: The Traditional Gap-Overlap Task (GOP)

### 3.1. Methods

#### 3.1.1. Participants

Since the experiment required young children to have certain visual attention and comprehension abilities, only high-functioning ASC children were included in the study. Other sampling criteria in Experiment 2 were the same as in Experiment 1. A different group of 20 Chinese ASC young participants (1 female, M_age_ = 5.92) and 25 age-matched Chinese TD children (2 females, M_age_ = 5.77) participated in Experiment 2. We conducted a similar power analysis for this experiment (parameter settings were the same as Experiment 1), and a total sample of at least 24 children is needed for the RM-ANOVA with the group as the between-group factor, and both task condition (baseline, overlap) and stimulus type (CI, non-CI) as within-group factors.

As with Experiment 1, all ASC children were previously diagnosed by more than one experienced clinician following the diagnostic criteria listed in the DSM-5 [1]. The AQ-Child scale [42] was also adopted to validate the clinical diagnosis, and children with ASC scored significantly higher on the AQ than their TD peers, *t* = 7.82, *p* < 0.001. IQ was evaluated using the Chinese version of Wechsler Preschool and Primary Scale of Intelligence-Fourth Edition (WPPSI-IV) [48], and there were no significant differences in verbal IQ (VIQ), performance IQ (PIQ) and full-scale IQ (FSIQ) between groups, |*t*| < 1.96, *p* > 0.05. Written consent was obtained from the participants’ parents before the formal experimental procedure. Before the formal experiment, all children participating were asked whether they wanted to play a game called “finding the red cross-hair”. If the child agreed, they would participate in the experiment(s). Detailed descriptions of participant demographics are presented in Table 3.

#### 3.1.2. Materials and Task

The traditional GOP task was utilized in Experiment 2. In the baseline condition, the central stimulus disappeared simultaneously with the onset of the target. However, in the overlap condition, the central stimulus continued to be present after the target’s appearance. Previous findings [20,21] with eye movement evidence have shown that the time needed to disengage from the central position is longer for the overlap condition compared to the baseline condition. This difference has been adopted to reflect endogenous disengagement processes, as participants must intentionally disengage attentional focus from the centrally presented stimuli to shift attention to the peripheral target in the overlap condition. 

In Experiment 2, the same 40 cartoon images (20 CI and 20 non-CI objects) used in Experiment 1 were adopted as the central stimulus in the GOP task, and each measured 150 × 100 pixels. The target object was a red crosshair and measured 50 × 50 pixels (approximately 1° in visual angle). This was presented either on the display screen’s left or right side at an eccentricity of 10° from the screen center. 

#### 3.1.3. Apparatus

Since Experiment 2 required fast saccadic responses to a target, rather than the free viewing paradigm used in Experiment 1, a higher sampling rate eye-tracker was required to capture participants’ oculomotor behaviors more sensitively. Therefore, in Experiments 2 (and 3), eye movements were recorded using the EyeLink Portable Duo eye-tracker (SR Research Ltd., Ottawa, Canada) with a 500 Hz sample rate. A 15.6-inch DELL screen (1920 × 1080 pixels, 60 Hz refresh rate) was utilized to display the stimuli. Children were seated approximately 52 cm away from the center of the screen, and a chinrest was utilized to maintain head stability during the formal eye movement testing session.

#### 3.1.4. Procedure

Firstly, a three-point calibration was performed to record the position of participants’ eyes (with a mean error below 0.5° acceptable for each child). Next, a one-point drift correction fixation target was presented at the center of the screen before every trial, and participants were instructed to look directly at this. Each trial began with a black crosshair (50 × 50 pixels) for 600–1000 ms to ensure participants were fixating on the center of the display at each trial start position. Following that, a central stimulus, which belonged to a CI or a non-CI item category, was presented for 1500–2000 ms. The target (a red crosshair) was then shown at an eccentricity of 10° on either the left or right side of the display screen for 2000 ms. In line with Fischer’s [19,20] studies, the central stimulus either disappeared (baseline condition) or was kept visible (overlap condition) during the target presentation. The children were instructed to watch the black crosshair first and then fixate on the central stimuli (CI or non-CI items). When the red crosshair (target) appeared, they were told to fixate on it. See Figure 3 for a trial sequence example of the classic GOP task. 

The experiment was divided into two blocks. One block contained 40 baseline trials and the other contained 40 overlap trials. Both blocks were performed following an ABBA order across participants. Each participant completed a total of 80 trials in a pseudo-randomized order.

#### 3.1.5. Eye Movement Measures and Data Analysis

Three thresholds were adopted for saccade detection with the following parameter settings: (1) the saccade velocity threshold was set at 30°/s; (2) the saccade acceleration threshold was set at 8000°/s^2^, which is recommended for cognitive research; (3) the saccade motion threshold, used to delay the onset of a saccade until the eye has moved significantly, was set to 0.15° [49]. In Experiment 2, three saccadic measures were adopted, SRT (ms), disengagement cost (DC), which was calculated according to the formula “disengagement cost (ms) = overlap SRT − baseline SRT”, indicating the extra attentional effort during endogenous disengagement [24], and failure rate (the proportion of the trials where participants did not shift their attention from the central stimuli to the peripheral target for the trial duration divided by the total number of valid trials in the overlap condition). Participants with higher DC or more failure rates meant they were less efficient in endogenous disengagement from the centrally presented stimuli [24,50].

Exclusion Criteria. Trials were excluded according to the following criteria: (1) trials contained a blink for the first saccade (4.40%), (2) the amplitude of the first saccade was less than 2° (9.47%) [51], (3) saccade positions exceeded 1° away from the center before the target onset (4.90%), (4) the saccade was initiated towards the opposite side of the target direction (0.92%), (5) SRT was beyond the range of 80–1000 ms (1.31%) [31,52], (6) SRT was greater or lower than three standard deviations away from the mean value of each participant (1.56%). The number of valid SRT trials included in the formal analysis was 2782.

Data were analyzed in RStudio (version 1.4.1717). The SRT was analyzed using the LMM method from the lme4 package (version 1.1-27.1) [47]. The fixed factors were group (ASC, TD), stimulus type (CI, non-CI), and task condition (baseline, overlap) in the LMM. In addition, the crossed random effects were specified for participants and trials. Since there was a limited number of data points for each participant’s DC score and a relatively low number of failure rates overall, these two measures were analyzed using the traditional ANOVA method. The RM-ANOVA was conducted to analyze the DC and failure rates, with the group (ASC, TD) as a between-group factor and stimulus type (CI, non-CI) as a within-group factor. Absolute values of the *t*-value equal to or greater than 1.96 indicate a significant difference.

### 3.2. Results

#### 3.2.1. Saccade Reaction Time (SRT) and Disengagement Cost (DC)

LMM analysis results showed a significant effect of condition (*b* = 47.84, *SE* = 4.20, *t* = 11.39, *p* < 0.001). SRT in the overlap condition was significantly longer than in the baseline condition (overlap SRT = 260 ± 127 ms; baseline SRT = 226 ± 73 ms). The effect of group, stimulus type, as well as all interactions did not reach significance (|*t*| < 1.96, *p* > 0.05). With respect to DC, no significant effect was observed for either group or stimulus type, and nor for the interactions of both factors (all *F*(1, 43) < 1.00, *p* > 0.05).

#### 3.2.2. Failure Rate

The measure of failure rate was not normally distributed (all *p*-values of Shapiro-Wilk < 0.001). Therefore, all data were square-root transformed during the analyses [28]. When comparing the failure rates between CI-related and non-CI-related items in both groups, the effect of group, stimulus type, and the group by stimulus type interaction were not significant (all *F*(1, 43) < 2.18, *p* > 0.05). Notably, the failure rate value was minimal for each condition in both groups. See Table 4 for the means and standard deviation of all measures in Experiment 2.

### 3.3. Summary

Experiment 2 investigated attentional disengagement ability by manipulating the foveal stimulus type (CI, non-CI) in ASC and TD groups, using the traditional GOP task and the same stimuli from Experiment 1. In contrast to our predictions, Experiment 2 found no group differences for the endogenous (overlap condition) or exogenous (baseline condition) disengagement processing from the CI-related and non-CI-related stimuli on all eye movement measures. Instead, both groups showed a similar attentional disengagement pattern unaffected by the stimulus type presented at the center of the display.

In line with prior research [19,20,21,53], we found the expected effect that the SRT was significantly longer in the overlap condition compared to the baseline condition across both ASC and TD groups. The presence of the foveal stimulus in the overlap condition requires more effort to voluntarily disengage attention from and move the eyes to the peripheral target. Traditionally, the GOP includes three task conditions: gap, baseline, and overlap [24,25], however in the current study, we did not include the gap condition because the temporal blank gap (e.g., 200 ms) between the presentation of the central object and the presentation of the peripheral target could have provided an apparent alert cue for the target appearance. Including a gap condition also results in the absence of a clear saccade initiation position prior to target presentation and can also lead to increased variability across eye-movement measures [14,20]. Including the baseline condition allowed us to measure the lag in endogenous disengagement in the overlap condition. It could, thus, reveal participants’ exogenous disengagement without the confounding effects on eye movements from obvious alert cues that might be brought about by a gap condition [7,21,24]. 

The findings related to delayed attentional disengagement in young children with ASC are controversial [24,25], and the findings from Experiment 2 provide no support for disengagement differences in ASC children. However, the failure in Experiment 2 to observe any disengagement differences between the two groups could be related to the potential effects of the adopted experimental conditions or task demands [54]. In the traditional GOP task, the central stimuli are presented for a duration of 1–2 s before the target’s appearance (e.g., [18,19,20,21,22]), during which participants might have fully processed the stimuli and prepared the next intentional eye movement. This experimental setup could make endogenous disengagement much easier and mask any existing group differences. In a recent study that adopted a different paradigm, the remote distractor paradigm (RDP), Zhang et al. [51] found that central stimuli presented as distractors resulted in a longer delay in initiating an eye movement to a simultaneously presented peripheral target in young children with ASC. In the RDP, on some trials, a central distractor item is presented with a peripheral target synchronously. In that condition, participants are required to ignore the distractor and make a saccade toward the target. In that paradigm, participants are required to respond as fast and as accurately as they can, so they do not have any time to prepare an eye movement before the presentation of the peripheral target. Therefore, a higher level of voluntary attentional control is required to disengage from centrally presented distractor items in the RDP, compared to the traditional GOP setup. This experimental difference could be a critical factor influencing whether differences between ASC and TD groups are observed in endogenous disengagement performance. It has been established that attentional control ability can be much lower in ASC, especially in young children with ASC [51]. 

Experiment 3 aimed to use a modified GOP paradigm (MGOP), based on the findings from the previously reported RDP paradigm, to examine further attentional disengagement characteristics for CI or non-CI-related objects (using the same stimuli from Experiment 1) in young children with ASC. In the MGOP task, we adopted similar baseline and overlap conditions as those used in Experiment 2. However, the central stimuli were shown only in the overlap condition, and presented synchronously with the peripheral target. This modification removes the extra time for young participants to pre-process the central stimuli in either the baseline or overlap condition, as in Experiment 2. In Experiment 3, it was predicted that participants would have to increase voluntary attentional efforts to disengage from the central stimuli in the modified overlap condition. Moreover, since previous research using an anti-saccade paradigm [38] has reported that ASC participants are less able to control attention when confronted with CI-related items, showing more involuntary saccadic errors to this kind of stimuli, we also predicted that the ASC group in Experiment 3 would have greater difficulties in voluntary disengagement processing relative to the TD group and that this effect would be more robust for the CI-related stimuli.

## 4. Experiment 3: The Modified Gap-Overlap Task (MGOP)

### 4.1. Methods

#### 4.1.1. Participants

The sampling criteria in Experiment 3 were the same as in Experiment 2. A new group of 20 Chinese ASC children (3 females, M_age_ = 5.54) and 24 age-matched Chinese TD children (4 females, M_age_ = 5.75) participated in Experiment 3. The power analysis results for the current study were extremely similar to those for Experiment 2, indicating that a total sample of at least 24 participants was needed for Experiment 3. Children with ASC were officially diagnosed following the DSM-5 [1] diagnostic criteria. The AQ-Child scale [42] was used to validate the clinical diagnosis, and children with ASC scored significantly higher on the AQ compared to the TD children, *t* = 6.40, *p* < 0.001. IQ was evaluated using the WPPSI-IV [48], and both groups were matched on VIQ, PIQ and FSIQ, |*t*| < 1.96, *p* > 0.05. Written consent was obtained from parents before taking part in the experimental procedures. Before the formal experiment, all children participating were asked whether they wanted to play a game called ”finding the star”. If the child agreed, they would participate in the experiment(s). See Table 5 for detailed demographic information for both groups. 

#### 4.1.2. Materials and Task

Two different conditions (improved baseline and overlap conditions) were included in the new MGOP paradigm. In the improved baseline condition, a central cross was first presented on which the participant was required to fixate. This was followed by a lateral target presented at an eccentricity of 10° on either the left or right side of the display screen in random order. In the improved overlap condition, the central stimulus (CI, non-CI) and the lateral target appeared simultaneously after fixation on the central cross. The same 40 cartoon objects (20 CI and 20 non-CI images) were adopted in Experiment 3 as the central stimuli in the overlap condition, each with a size of 300 × 200 pixels. The peripheral target was a star set at a size of 50 × 50 pixels (approximately 1° in visual angle). 

#### 4.1.3. Apparatus

In line with Experiment 2, eye movements were recorded using the EyeLink Portable Duo eye-tracker (SR Research Ltd., Ottawa, ON, Canada) with a sampling rate of 500 Hz. All images were displayed on a 15.6-inch DELL screen (1920 × 1080 pixels) with a refresh rate of 60 Hz. Children were invited to sit at a distance of approximately 52 cm from the screen, and a chin rest was used to maintain head stability.

#### 4.1.4. Procedure

Firstly, a three-point calibration was performed, and a mean error below 0.5° was accepted for each child. Each trial began with a one-point drift correction whereby a dot was presented at the center of the screen to attract the participant’s attention. Next, a red crosshair (50 × 50 pixels), which had to be fixated on, appeared for 600–1000 ms. Following that, a target star was shown in isolation (baseline condition) or was presented with the central CI-related or non-CI-related item (overlap condition). Next, targets were presented either on the left or right side of the display at an eccentricity of 10° away from the center for 2000–3000 ms. Finally, a 500 ms blank screen appeared. Children were instructed to fixate on the red crosshair first and then look at the target (star) on trials where it appeared. See Figure 4 for a trial procedure example of the MGOP task. As with Experiment 2, a total of 80 trials presented in a pseudo-randomized order were included in the current experiment. The trials were divided into two blocks of baseline and overlap conditions which were shown to participants following an ABBA design. 

#### 4.1.5. Eye Movement Measures and Data Analysis

The saccadic parameter settings in Experiment 2 were the same as in Experiment 3. Moreover, in line with Experiment 2, we took SRT (ms), DC (ms), and failure rate as measures for analysis in Experiment 3. We first compared the group differences for the baseline and the overlap conditions. Then, the influence of central stimuli type on the disengagement process was compared using only the overlap trials in the analysis. 

Exclusion Criteria. SRT trials were excluded according to the following criteria: (1) trials contained a blink for the first saccade (6.92%), (2) the amplitude of the first saccade to the target was less than 2° (16.67%) [51], (3) saccade positions exceeded 1° away from the center before the target onset (2.59%), (4) the saccade was initiated towards the opposite side of the target direction (1.03%), (5) SRT was beyond the range of 80–1000 ms (1.74%) [31,52], (6) SRT was greater or lower than three standard deviations away from the mean value of each participant (0.91%). After data filtering, 2462 valid SRT trials remained for further analysis.

The LMM method was adopted to analyze the SRT in RStudio (version 1.4.1717; lme4 package version 1.1-27.1) [47]. In the SRT by condition analysis, the fixed factors were group (ASC, TD) and condition (baseline, overlap). In the SRT by stimuli type analysis, the fixed factors were group (ASC, TD) and stimulus type (CI, non-CI). The crossed random effects were specified over participants and trials in both analyses. In line with Experiment 2, a repeated measures ANOVA was then conducted to analyze the DC and failure rate, with a group (ASC, TD) as a between-group factor and stimulus type (CI, non-CI) as a within-group factor. Absolute values of the *t*-value equal to or greater than 1.96 indicate a significant difference.

### 4.2. Results

#### 4.2.1. Saccade Reaction Time (SRT) and Disengagement Cost (DC)

SRT by condition LMM analysis: Results showed significant effects of group (*b* = −69.98, *SE* = 15.15, *t* = −4.62, *p* < 0.001) and condition (*b* = 232.37, *SE* = 4.66, *t* = 49.91, *p* < 0.001). Children with ASC displayed a longer SRT than TD children for the group effect. For the condition effect, the SRT in the overlap condition was significantly longer than that in the baseline condition. Moreover, the group by task interaction was also significant (*b* = −99.36, *SE* = 9.29, *t* = −10.70, *p* < 0.001). Further analysis showed that children with ASC demonstrated a delayed SRT to disengage from the foveal stimuli compared to their TD peers in the overlap condition (*b* = −138.30, *SE* = 35.77, *t* = −3.87, *p* < 0.001). However, no group difference was found in the baseline condition (|*t*| < 1.96, *p* > 0.05). See Figure 5a for mean SRT by task condition in each group.

SRT by stimulus type LMM analysis: A significant main effect of group (*b* = −121.21, *SE* = 34.56, *t* = −3.51, *p* = 0.001), indicated that children with ASC needed longer to disengage from foveal objects compared to their TD peers. However, the main effect of the central stimulus type and the group by stimuli type interaction was not significant (|*t*| < 1.96, *p* > 0.05). See Table 6 for the means and standard deviation of SRT in the MGOP task and Table 7 for more details from the LMM analysis.

DC analysis: We compared DC across the two foveal stimulus types between ASC and TD groups. This analysis removed two ASC children due to a lack of valid data in the overlap condition. A significant main effect of group was found (*F*(1, 40) = 12.02, *p* = 0.001, *η*^2^ = 0.22), indicating that children with ASC (292 ± 133 ms) had a higher DC to disengage from central items compared with TD children (180 ± 77 ms). The main effect of stimulus type and the interaction of both were not significant (*F*(1, 40) < 1.00, *p* > 0.05). 

#### 4.2.2. Failure Rate

Since the failure rate was not normally distributed (all *p*-values of Shapiro-Wilk < 0.001), all data were square-root transformed during the analyses [28]. A significant main effect of group (*F*(1, 42) = 20.55, *p* < 0.001, *η*^2^ = 0.28) indicated that the failure rate was significantly greater in the ASC group than in the TD group. The main effect of stimulus type was also significant (*F*(1, 42) = 14.19, *p* < 0.001, *η*^2^ = 0.03), suggesting that children had higher failure rates in the CI-related conditions than in the non-CI conditions. This finding was qualified by a group by stimulus type interaction (*F*(1, 42) = 14.49, *p* < 0.001, *η*^2^ = 0.03) which showed that children with ASC had higher failure rates for CI-related items compared with non-CI items (*t* = 5.13, *p* < 0.001). The TD group rarely showed disengaging failures across the two types of stimuli (|*t*| < 1.96, *p* > 0.05). See Figure 5b for the mean failure rate for each group across two stimulus types. 

#### 4.2.3. Saccade Reaction Time (SRT) Differences between Experiment 2 and Experiment 3

We combined the SRT data from both Experiment 2 and Experiment 3 (including 40 ASC and 49 TD young participants; 5244 trials in total) to compare the traditional GOP task (Experiment 2) with our modified GOP version (Experiment 3). We used LMM analyses where the fixed factors were experiment (Experiment 2, Experiment 3), group (ASC, TD), and condition (baseline, overlap). There was a significant interaction between the three fixed factors of experiment, group and condition (*b* = −91.62, *SE* = 10.68, *t* = −8.58, *p* < 0.001). Further contrasts showed that the overlap SRT of Experiment 3 in the ASC group was significantly longer than the overlap SRT in Experiment 2 for both ASC (*b* = −223.88, *SE* = 17.00, *t* = −13.18, *p* < 0.001) and TD groups (*b* = −219.22, *SE* = 16.10, *t* = −13.59, *p* < 0.001). In addition, the baseline SRT was relatively steady across different experimental conditions and participant groups (|*t*| < 1.96, *p* > 0.05). These results reveal an intact exogenous disengagement performance in ASC young children across different experiments and verify that the MGOP task is a better paradigm to reveal actual endogenous attentional disengagement differences between the ASC and TD groups. See Figure 6 for the mean SRT of each group in both Experiment 2 and Experiment 3.

### 4.3. Summary

Experiment 3 employed an MGOP task to investigate whether there were attentional disengagement differences, for CI and non-CI items, in children with and without ASC. Compared to the TD group, young children with ASC generally exhibited slower, less efficient endogenous disengagement (delayed SRT, greater DC, and more failure rates in the overlap condition). Furthermore, CI-related items produced higher failure rates than non-CI-related items in the ASC group, although this measure had high variability. An inspection of the data included in the failure rate analysis showed that 14 out of the 20 ASC children produced failure rates, with most of these ASC children producing 1–5 failure trials and two participants producing equal or over 10 failure-to-move trials. Accordingly, this measure provided some evidence of disengagement difficulties for CI-related items. However, the failure rate trials were small in number, and there were clear individual differences within the ASC group on this measure. Similar to Experiment 2, no difference in exogenous disengagement was found between the two groups (similar SRT in the baseline condition) in Experiment 3. Overall, this experiment has confirmed that young children with ASC show evidence for visual disengagement problems, and the presence of CI-related items at fixation may aggravate this characteristic for some ASC children. The hypothesis of decreased attentional disengagement, which may be one of many primary characteristics associated with attentional processing in ASC, was supported in the current study.

In contrast with the findings from Experiment 2, the findings from Experiment 3 supported our initial expectations. Compared to the results of Experiment 2, the results from Experiment 3 suggested: (1) evidence for significant group differences in endogenous attentional disengagement processing and (2) stimulus types modulated disengagement performance in the ASC group. These observed differences can be attributed to the new MGOP paradigm. However, they are unlikely to result from altered arousal regulation in ASC since both the modified overlap condition (sudden appearance of the central stimuli) and the baseline condition (sudden disappearance of the foveal fixation) could act as warning signals to all participants in both groups. Under the overlap condition in this modified paradigm, where participants do not have an opportunity to pre-process the foveal stimulus (as is the case in the standard GOP paradigm), we have found evidence for disengagement costs for the baseline condition compared to the overlap condition in both groups, and SRT differences in attentional disengagement between the two groups. Further evidence supporting adopting the MGOP paradigm to investigate attentional disengagement differences is apparent if we look at the SRT for the overlap condition in Experiment 2 (260 ± 127 ms) compared to those in Experiment 3 (397 ± 148 ms). Additionally, the CI effect on ASC was shown to measure failure rates, another crucial measure in visual disengagement studies [10,14,15,55], which reflects an inability to disengage from the foveal engaged stimuli and hence a failure to look to the peripheral target [28]. This immersive processing pattern for CI items rarely occurred in the ASC group in Experiment 2. This demonstrates that the traditional GOP may not be suitable for investigating group or complex stimulus differences in disengagement.

Notably, the evidence from the current study could explain why numerous autistic studies did not find any group differences in attentional disengagement (e.g., [19,20,31]). For instance, Fischer et al. [20] (who presented foveal stimuli for 2000 ms) found no sign of delayed attentional disengagement in young toddlers with ASC, either in SRT or in the overall rate of successful disengagement. It is potentially possible that these children with ASC did not exhibit preserved attentional disengagement. However, the results could also reflect pre-processing of central stimuli prior to target presentation, similar to the observations from Experiment 2 in the current study. In Fischer’s studies, less effort was required to disengage from the central stimuli regardless of stimulus types for both participant groups. In comparison, the new overlap condition does not provide any time to pre-process the foveal stimuli prior to target presentation, and this modulation results in increased SRT to the target in both groups. 

The findings from Experiment 3 align with Landry and Bryson [15], who examined attentional disengagement performance in young children with ASC and found similar exogenous disengagement latency, but impaired endogenous attentional disengagement with a significant delayed SRT and more failure rates compared with TD children. In addition, in line with previous reports [5,35,39], the current experiment provided evidence for the modulation of disengagement behavior, specifically for cartoon CI-related foveal stimuli in some children in the ASC group.

## 5. General Discussion

In this systematic study, using the same cartoon materials, we conducted three eye-tracking experiments to investigate attentional engagement and disengagement differences across two stimulus types (CI, non-CI related items) in young Chinese children with and without ASC. In Experiment 1, we aimed to test attentional allocation for the two stimuli types in a visual preference task, and we found a bias to attend to CI-related items in the ASC group. The CI-related items attracted and held the attention of ASC children relative to non-CI items, and this preference was absent in TD children. Following the clear evidence from Experiment 1, we explored whether the same stimuli influenced ASC children’s attentional disengagement processing. Experiment 2 aimed to investigate whether the attentional preference for CI-related items observed for the ASC children in Experiment 1 would influence the nature of attentional disengagement in a classic GOP task. This was not the case: both groups showed a similar disengagement effect, regardless of the stimulus type. However, this null group difference was thought to have resulted from the specific experimental setup in Experiment 2 rather than an absence of any group disengagement differences, which had allowed for pre-processing of the central stimuli in both baseline and overlap conditions prior to the presentation of the target [51,54]. Consequently, for Experiment 3, we devised a modified version of the paradigm (MGOP), which removed the opportunity to pre-process the foveal items before the target display onset. As predicted, Experiment 3 provided data to suggest that the original GOP task was unsuitable for eliciting voluntary disengagement differences from groups and stimuli. The findings also showed that young ASC children exhibited visual disengagement problems compared to their TD counterparts. Furthermore, Experiment 3 showed that attentional disengagement processing was influenced more in the presence of CI-related stimuli for a proportion of the ASC children. Performance for exogenous disengagement was similar between ASC and TD groups across Experiments 2 and 3.

A clear finding from the current study was that young children with ASC showed an attentional preference to engage with CI-related materials. Circumscribed interest is a prominent feature of non-social symptoms within the ASC diagnostic factor of repetitive behaviors, and CI is observed in most children with ASC [32,39]. The CI-related attentional experience could give ASC children a sense of pleasure and reward, and such feedback is associated with more activation in neural reward circuitry [36,39,56]. Attentional preference patterns for authentic CI items can characterize ASC from a young age [5,37], and we found this similar characteristic for our cartoon CI images in Experiment 1. For preschoolers, cartoon materials fit with their aesthetic experiences, and these materials (e.g., picture books) frequently appear in their learning surroundings. While young TD children might be able to effectively process critical messages (especially social information) from cartoon materials in daily learning, young ASC children might be more immersed in CI-related items (e.g., trains). As a result, they may be less likely to acquire valuable social information [5,35,36,37,39]. It could also be argued that this increase in engagement with CI items in ASC could impact attentional disengagement processing and negatively influence the development of core social interactive skills (e.g., joint attention) [6,9,24].

However, in line with recent related studies that have employed the classic GOP task (e.g., [18,19,20,21,31]), we found no evidence of differences in attentional disengagement across the two stimulus types in the ASC and TD groups in Experiment 2. In the classic GOP, participants are presented with the central stimulus for a duration of 1500–2000 ms on the first display screen, which is then followed by the display that has the target presented in the periphery, which is presented in isolation (baseline) or with the same central stimulus from the previous display (overlap). Thus, participants may need less effort to process the item again when it is presented with the target since they already had sufficient time to do this before the target display. In essence, the different stimulus types presented in the second display can be ignored equally well by both participant groups and for both stimulus types. This argument can apply to the previous studies that also failed to find group disengagement effects using the classic GOP [18,19,20,21,30,31].

In order to avoid the confounding effects of pre-processing the central stimuli, we removed the chance to do this in both the baseline and the overlap conditions in our MGOP task. This permitted us to observe effects on the eye movement system, such as attentional holding by the item. In Experiment 3, young children with ASC showed a general delayed reaction and higher failure rates to disengage compared to their TD peers in the overlap condition. The hypothesis that delayed disengagement might be one of the primary attentional characteristics associated with ASC [6] was confirmed in this study. Moreover, in the ASC group, endogenous disengagement performance was influenced more by the presence of CI-related items. In addition to these findings, we found another interesting phenomenon in relation to attentional processing in ASC: disengagement differences across stimuli types were only reflected in the failure rate measure. In other words, the appearance of CI-related objects had a negligible effect on disengagement speed (e.g., SRT). However, it significantly impacted the ability to ignore the CI-related objects and make an eye movement toward the target. Together, these measures of SRT and failure rate suggest the following: (a) in the trials where the ASC children could successfully endogenously disengage from the central objects, differences between stimulus types did not affect their dynamic saccade behaviors, and (b) higher failure rates to CI-related items could reflect reduced cognitive control of visual attention in the presence of CI items [38]. The negligible effect on disengagement speed might also signify that the best way to evaluate attentional deficits in ASC using eye movements may be to focus on disengagement using paradigms that better reflect real-world scenarios rather than utilizing low-level eye movement paradigms where the task demands do not allow for significant engagement in the first place.

Limitations and Future directions. In the current study, static stimuli were adopted across all three experiments. Future studies could investigate whether this deviant attentional pattern in ASC is more severe under dynamic conditions. This would have greater ecological validity and might better reflect how information is attended to and processed in the real world. Furthermore, the novel cartoon stimuli used in the study have provided robust evidence that they can elicit engagement and, to some extent, disengagement biases in children with ASC. Since these kinds of stimuli are readily available in everyday tasks such as watching cartoon videos, reading picture books, etc., future work could be carried out to focus on their impact on the development of typical cognition and on the development of rehabilitation materials to improve attention in ASC. Additionally, the related neural mechanisms that operate during attentional disengagement in children with ASC should also be explored to highlight whether group differences at the neural level reflect the behavioral effects observed in the current study.

## 6. Conclusions

The current study reports novel findings from experiments where the same cartoon materials are used in classic and modified eye-movement paradigms to show that Chinese children with ASC engage more with CI-related items (when these are presented with non-CI-related items). They disengage less quickly from centrally presented items compared to TD peers overall, and they disengage less frequently from centrally presented CI related items compared to non-CI related items (when required to saccade to a peripheral target). These effects reflect attentional processing differences in the endogenous attentional system in ASC, which is known to mature later than the exogenous system [27]. Additionally, the findings highlight that adopting classic low-level paradigms to investigate disengagement processes in ASC might mask rather than reveal any group differences. They explain the inconsistencies reported in the literature in relation to reports in this area from classic GOP studies. Furthermore, the finding of a propensity to remain engaged with whatever is being currently ‘looked at’ (Experiment 1) coupled with a failure to disengage from CI-related stimuli (Experiment 3) has consequences for cognitive development in ASC. The overall findings of the ASC differences in attentional engagement and disengagement have clear implications for the development of typical social cognition since a bias to over-attend to some stimuli coupled with a bias to show increased disengagement for the same stimuli would prevent the detection and opportunity to respond effectively to signals or events in the current environment. These attentional processing differences could potentially influence joint attention [9], spoken word acquisition [10], and emotion processing [8], all of which belong to diagnostic characteristics in the communication domain in ASC.

## Figures and Tables

**Figure 1 brainsci-12-01461-f001:**
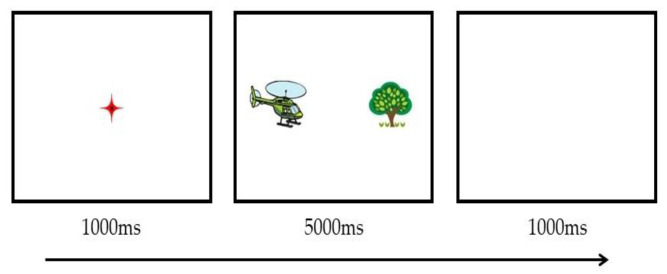
A schematic example of a trial sequence in Experiment 1. The airplane on the left is an example of a CI-related item, and the tree on the right is a non-CI item.

**Figure 2 brainsci-12-01461-f002:**
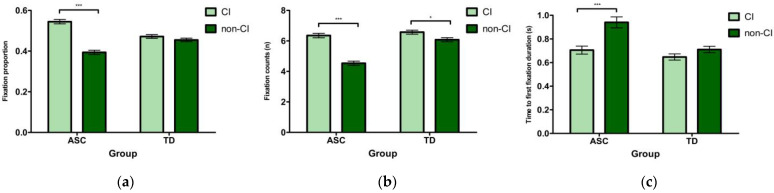
Analysis results related to three eye movement measures for both groups in Experiment 1. (**a**) Preference, (**b**) Detail Orientation, (**c**) Prioritization. Error bars represent standard errors. * *p* < 0.05, *** *p* < 0.001.

**Figure 3 brainsci-12-01461-f003:**
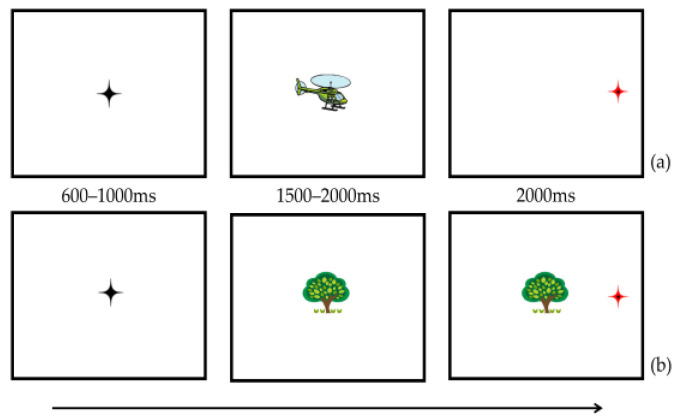
Schematic examples of the GOP task in the baseline condition (**a**) and overlap condition (**b**) of Experiment 2. The helicopter represents a CI-related item, and the tree represents a non-CI item.

**Figure 4 brainsci-12-01461-f004:**
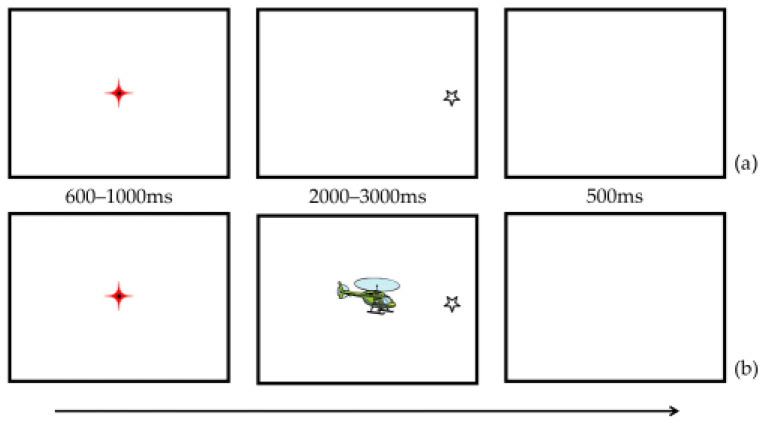
Schematic examples of the MGOP task in the baseline condition (**a**) and overlap condition (**b**) of Experiment 3. The helicopter picture represents a CI-related item.

**Figure 5 brainsci-12-01461-f005:**
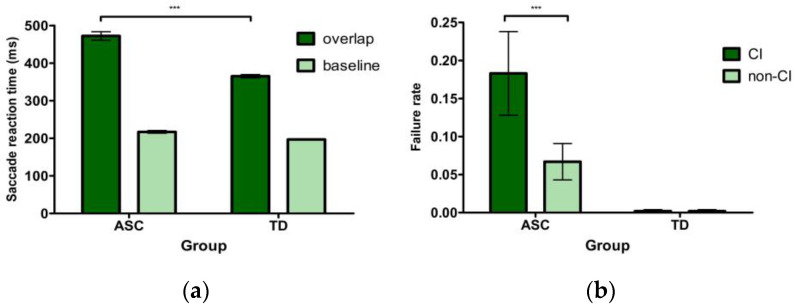
Mean eye movement measures for each group in Experiment 3. (**a**) saccade reaction time by condition, (**b**) failure rate. Error bars represent stand errors. *** *p* < 0.001.

**Figure 6 brainsci-12-01461-f006:**
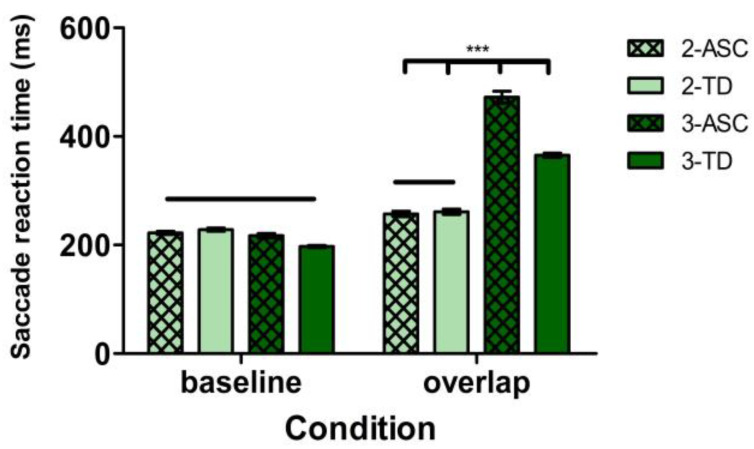
Mean SRT for each group in both Experiment 2 and Experiment 3. Error bars represent standard errors. *** *p* < 0.001.

**Table 1 brainsci-12-01461-t001:** Demographics (mean ± SD) of the ASC and TD groups in Experiment 1.

	ASC (*n* = 22, 2 Females)	TD (*n* = 22, 2 Females)	*t*	*p*
age (in years)	4.95 (0.59)	5.14 (0.44)	−1.20	0.236
verbal-IQ	87.00 (25.35)	95.59 (20.30)	−1.24	0.222
AQ-Child	86.00 (9.39)	57.77 (10.34)	9.23	<0.001

ASC—autism spectrum condition; TD—typically developing; AQ—autism quotient.

**Table 2 brainsci-12-01461-t002:** Fixed effect estimates for three measures in Experiment 1.

	Preference			Detail Orientation			Prioritization		
	*b*	*SE*	*t*	*b*	*SE*	*t*	*b*	*SE*	*t*
group	−0.01	0.01	−0.60	0.91	0.25	3.58 ***	−0.14	0.04	−3.44 **
stimulus type	−0.08	0.01	−8.59 ***	−1.13	0.14	−8.34 ***	0.17	0.04	4.35 ***
group × stimulus type	0.13	0.02	6.83 ***	1.34	0.27	4.92 ***	−0.22	0.08	−2.75 **
ASC: CI vs. non-CI	−0.15	0.02	−10.09 ***	−1.80	0.20	−9.18 ***	0.28	0.06	4.31 ***
TD: CI vs. non-CI	−0.02	0.01	−1.34	−0.47	0.19	−2.48 *	0.06	0.05	1.34

ASC—autism spectrum condition; TD—typically developing; CI—circumscribed interest; * *p* < 0.05, ** *p* < 0.01, *** *p* < 0.001.

**Table 3 brainsci-12-01461-t003:** Demographics (mean ± SD) of the ASC and TD groups in Experiment 2.

	ASC (*n* = 20, 1 Female)	TD (*n* = 25, 2 Females)	*t*	*p*
age (in years)	5.92 (1.13)	5.77 (0.77)	0.53	0.597
VIQ	106.55 (18.74)	105.20 (10.08)	0.31	0.759
PIQ	105.85 (15.49)	108.96 (17.32)	−0.63	0.534
FSIQ	107.00 (16.03)	105.08 (11.27)	0.47	0.640
AQ-Child	80.25 (17.17)	48.28 (9.97)	7.82	<0.001

ASC—autism spectrum condition; TD—typically developing; VIQ—verbal IQ; PIQ—performance IQ; FSIQ—full-scale IQ; AQ—autism quotient.

**Table 4 brainsci-12-01461-t004:** The means and standard deviation of all measures in Experiment 2.

	ASC		TD	
	CI	non-CI	CI	non-CI
baseline SRT (ms)	219 (62)	225 (72)	230 (80)	227 (76)
overlap SRT (ms)	258 (122)	256 (119)	263 (130)	261 (133)
DC (ms)	42 (62)	33 (54)	33 (61)	32 (72)
failure rate(pre-transformation)	0.017 (0.030)	0.021 (0.039)	0.008 (0.027)	0.010 (0.028)

ASC—autism spectrum condition; TD—typically developing; CI—circumscribed interest; SRT—saccade reaction time; DC—disengagement cost.

**Table 5 brainsci-12-01461-t005:** Demographics (mean ± SD) of the ASC and TD groups in Experiment 3.

	ASC (*n* = 20, 3 Females)	TD (*n* = 24, 4 Females)	*t*	*p*
age (in years)	5.54 (0.95)	5.75 (0.52)	−0.95	0.350
VIQ	107.90 (15.74)	106.50 (7.19)	0.39	0.698
PIQ	110.10 (14.50)	114.33 (13.39)	−1.01	0.320
FSIQ	112.15 (12.57)	111.63 (9.00)	0.16	0.873
AQ-Child	80.85 (10.76)	56.00 (14.30)	6.40	< 0.001

ASC—autism spectrum condition; TD—typically developing; VIQ—verbal IQ; PIQ—performance IQ; FSIQ—full-scale IQ; AQ—autism quotient.

**Table 6 brainsci-12-01461-t006:** The means and standard deviation of saccade reaction time (ms) in Experiment 3.

Group	Condition	M (SD)	Group	Stimulus Type	M (SD)
ASC	baseline	217 (91)	ASC	CI	467 (176)
	overlap	472 (188)		non-CI	476 (197)
TD	baseline	197 (48)	TD	CI	362 (106)
	overlap	365 (114)		non-CI	369 (122)

ASC—autism spectrum condition; TD—typically developing; CI—circumscribed interest.

**Table 7 brainsci-12-01461-t007:** Fixed effect estimates on saccade reaction time in Experiment 3.

Saccade Reaction Time by Condition				
	*b*	*SE*	*t*	*p*
group	−69.98	15.15	−4.62	<0.001
condition	232.37	4.66	49.91	<0.001
group × condition	−99.36	9.29	−10.70	<0.001
baseline: ASC vs. TD	−20.14	10.93	−1.84	0.072
overlap: ASC vs. TD	−138.30	35.77	−3.87	<0.001
**Saccade Reaction Time by Stimulus Type**				
	*b*	*SE*	*t*	*p*
group	−121.21	34.56	−3.51	0.001
stimulus type	−4.26	10.28	−0.42	0.680
group × stimulus type	4.96	15.01	0.33	0.741

ASC—autism spectrum condition; TD—typically developing.

## Data Availability

Main data presented in this study are included in this article. Data can be found on the OSF at the following link: https://osf.io/ry8a5/ (accessed on 10 September 2022).

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
