# Peer review of "Attentional Engagement and Disengagement Differences for Circumscribed Interest Objects in Young Chinese Children with Autism"

_brainsci, 2022, doi:10.3390/brainsci12111461_

Round 1

Reviewer 1 Report (New Reviewer)

This is a really nicely presented paper, which is very clearly written and describes a series of experiments investigating oculomotor/ attentional engagement / disengagement in Autistic children with different types of objects as fixation stimuli. The schematics of the experimental tasks are really excellent and make the methods very easy to understand as are the results thanks to some very clear figures and tables. 

I only have one suggestion for the paper. I wasn't familiar with the idea of circumscribed interest objects. Please can more background be given on how CI versus non-CI objects are defined. In terms of interpreting the findings, is it possible CI objects are visually different to non-CI objects, such that more sensory properties drive the differences? Or is more to do with engagement of analytical/categorical processes during viewing of these objects which explains different processing in ASD?

Author Response

Reviewer 2 Report (Previous Reviewer 2)

1.      Describe the novelty of the article made by the author? From the results of my evaluation, it seems that many similar published works adequately explain what you have raised in the current manuscript. If there is something others really new in this manuscript, please highlight it more clearly in the introduction section.

2.      Previous research has to be explained in the introduction section, including their work, novelty, and limits, to illustrate the research gaps that will be filled in the current study.

3.      Please kindly keen about comments number 1 and 2. However, by condensing the substance and shortening the introduction.

4.      A previous study from Afif et al. that cooperated children with autism spectrum disorder using a hug machine needs to be explained and adopted in the manuscript. Afif, I. Y.; Manik, A. R.; Munthe, K.; Maula, M. I.; Ammarullah, M. I.; Jamari, J.; Winarni, T. I. Physiological Effect of Deep Pressure in Reducing Anxiety of Children with ASD during Traveling: A Public Transportation Setting. Bioengineering 2022, 9, 157. https://doi.org/10.3390/bioengineering9040157

5.      To help the reader grasp the study's workflow more easily, the authors could include more visuals to the materials and methods section in the form of figures rather than sticking with the text that now predominates.

6.      What is the basis for participant involved selection? Is there any protocol, standard, or basis that has been followed? It is unclear since the patient is very heterogeneous with a small number. The resonance involved impacts the present result makes this study flaws. One major reason for rejecting this paper.

7.      Outcomes must be compared to similar past research.

8.      What is the limitation of the present work? Please include it before the conclusion section.

9.      Further research should be discussed in the conclusion section.

10.   The reference should be enriched with literature from the last five years. Literature published by MDPI is strongly recommended.

11.   Due to grammatical problems and linguistic style, the authors should proofread the work. It would be used MDPI English editing service for this concern.

12.   Please ensure that the authors followed the MDPI format correctly; modify the current form and recheck, as well as any other problems that have been highlighted.

Author Response

Reviewer 3 Report (New Reviewer)

Summary

This manuscript reports results from an experimental study with Chinese autistic children. The study focused on the attentional processing differences between children with ASC and TD children when shown circumscribed and non-circumscribed related stimuli.

Thank you for the opportunity to review this manuscript. I enjoyed reading this manuscript and I think it has considerable potential to contribute to the literature in several ways. It can add further evidence to the literature on better understanding the characteristics of children ASC, how ASC further affects attentional processing skills, as well as provide further evidence on the use of gap-overlap paradigm, as there have been convergent findings so far.

Introduction

Very clear and detailed account of a wide range of studies on the research conducted around the attentional processing skills of children with ASC has been provided and it is a key advantage of this manuscript. I think the very detailed review of the studies on ASC, the ways it affects children’s attentional processing skills, and the detailed description of the versions of the GOP tasks significantly helped build the rationale for this study.

I think the use of sub-headings in this section would be beneficial in order to clearly show the relevant topic areas that have been covered in terms of previous literature. Rationale for this study has been presented well and the choices around the conditions for the GOP experiments for this study have been justified sufficiently.

Method

A detailed account of the sampling method and eye movement measures, and tools has been provided for each experiment, so that enhanced the clarity of the methods employed. I appreciated the inclusion of the power analysis on the recruited sample prior to data collection commencing. Some details to explain the rationale behind the decision to change the apparatus for eye-tracking measures between Experiment 1 (Tobii) and Experiments 2 and 3 (EyeLink) are needed.

It would be beneficial for authors to clarify how consent from the children themselves, who were participating in this study, was obtained, as there is only mention of parental consent.

Results

Very clear and detailed presentation of the statistical analyses for all three experiments and I think the inclusion of the results in figures helps highlight the key findings for the experimental groups and conditions. The structure of the sections for each experiment was not the one I was expecting, but it actually illustrates the results of each experiment very well in this way.

Discussion

The authors have provided a comprehensive description of the many results and findings from these three experiments as well as an explanation of their findings in the light of previous literature. The results provide an interesting contribution to the existing literature in terms of the application of the GOP task in assessing the attentional processing skills of children with ASC, as well as providing additional evidence to help better understand the impact of ASC on children’s cognitive skills. I think that some details around the limitations of the study as well as suggestions for further research on this topic would be useful as part of the general discussion.

Round 2

Reviewer 2 Report (Previous Reviewer 2)

Reviewers greatly appreciate the efforts that have been made by the author to improve the quality of their articles after peer review. I reread the author's manuscript and further reviewed the changes made along with the responses from previous reviewers' comments. Unfortunately, the authors failed to make some of the substantial improvements they should have made making this article not of decent quality with biased, not cutting-edge updates on the research topic outlined. In addition, the author also failed to address the previous reviewer's comments, especially on comments number 1, 2, 4, and 5. With all due respect, the reviewer opposed this article to be published and must be rejected. Thank you very much for the opportunity to read the author's current work.

This manuscript is a resubmission of an earlier submission. The following is a list of the peer review reports and author responses from that submission.

Round 1

Reviewer 1 Report

Thank you for the opportunity to review this interesting study. My general impression is positive – I think I will make a contribution to the field. At the same time, I have several questions regarding methodology and interpretation which should be addressed. Please see specific comments below.

Introduction

“However, individuals 48 with ASC are found to have difficulties in disengaging from fixated stimuli when they 49 make overt eye movements to new targets [7,14–16].” -> p2. 48-49. I think this is too strong. As the authors state later in the manuscript, a considerable number of studies (including those by Fisher et al) have found no evidence for disengagement difficulties in ASC.

I would suggest that the authors change the term “deficit” to a more inclusive term.

The description of the components in the GOP is largely correct, but in my opinion misses the potential role of the alerting network and phasic alerting. As noted in several previous papers (e.g., Jin & Reeves 2009, Vis. Res.; Kleberg et al, 2017 JADD, Kleberg et al., 2020, ECAP;  Reuter-Lorenz et al 1995, Exp. Brain Res.; Keehn, Müller & Townsend, 2013), the disappearance of the central stimulus in the overlap (and to some extent, the baseline) condition functions as a spatially non-specific warning cue which decreases reaction time speed and leads to a transient increase in arousal. Therefore, larger relative SRT differences between overlap/baseline and gap trials could partly be due to impaired arousal regulation and alterations in the alerting network. I would suggest the authors to include this in the discussion of the components measured by the GOP, especially as it may be relevant to the interpretation of results from experiment 3.

Participants

In all three experiments, sample size should be motivated. Especially in experiment 2, sample size is very small, and I am not convinced that a null effect is informative.

Was there any information about the level of CI in participants?

Data analysis

In all analyses, it is crucial to provide a detailed description of how eye tracking data were analyzed, especially about the fixation filter which was applied, and why it was chosen. Several metrics such as SRT and fixation count are sensitive to the choice of fixation filter and its parameters.

Stimuli

Did the authors validate that the stimuli in the CI category was indeed corresponding to the Cis of participants?

Statistics and results

Please report exact p-, t- and b-values for non-significant effects.

Please note that p can’t be zero.

In 2.2.2: the interaction effect should be followed up with pairwise contrasts since it’s difficult to interpret main effects in the presence of strong interactions.

In 4.2.3.: It seems unjustified to compare data from experiments 2 and 3 since they had non-overlapping samples. If I understand the analysis correctly, since no random effects for individual was included, the model basically treats each trial as coming from a unique individual which means pseudo-replication. Please change or clarify.

Experiment 4:  The authors find a strong effect of group in the overlap condition in their modified GOP task. My question is whether this effect could result from altered arousal regulation in the ASC group, since the sudden appearance of the central stimulus functions as a warning signal, OR if it could result in increased attention allocation to the (novel) central stimulus rather than disengagement.

Which instructions did participants get in the three experiments?

Reviewer 2 Report

Dear Zhou et al.,

The manuscript “Attentional Engagement and Disengagement Differences for Circumscribed Interest Objects in Young Chinese Children with Autism Spectrum Condition: An Eye Movement Study” (brainsci-1698532) by Zhou et al. aimed to investigate attentional processing differences for circumscribed interest (CI) and non-CI objects in young Chinese children with autism spectrum condition (ASC) and typically developing (TD) controls. The topic is interesting, but I think this article should reconsider after proper changes in major revision for publication in Brain Science. Some of my specific comments are below:

  1. The title of present manuscript (line 2-4) is too long which makes it harder to understand. It is recommended to shorten the tittle.
  2. In the abstract section (line 13-26), the authors should add quantitative results rather than only qualitative results.
  3. Describe the novelty of the article made by the author? From the results of my evaluation, it seems that many similar published works adequately explain what you have raised in the current manuscript related to eye movement studies in children with autism spectrum disorder based on the best reviewer knowledge in this research area. If there are something others really new in this manuscript, please highlight it more clearly in the introduction section (line 30-123).
  4. The state of the art and the significance of the current study are not clearly present, the authors should highlight it more advanced in the introduction section (line 30-123).
  5. What is the reason for Chinese children studied in the present article, nothing any explanation in the introduction section (line 30-123). It is so strange when the sample mention specifically “Chinese children” but nothing any clear explanation in the introduction.
  6. Since this manuscript is related to autism spectrum disorder, I would encourage and advise the authors to adopt some of the specific additional references related to autism spectrum disorder in the introduction section (line 30-123) as follow:
    • Physiological Effect of Deep Pressure in Reducing Anxiety of Children with ASD during Traveling: A Public Transportation Setting. Bioengineering 2022, 9, 157. https://doi.org/10.3390/bioengineering9040157
    • Effect of Short-Term Deep-Pressure Portable Seat on Behavioral and Biological Stress in Children with Autism Spectrum Disorders: A Pilot Study. Bioengineering 2022, 9, 48. https://doi.org/10.3390/bioengineering9020048
    • The Subjective Comfort Test of Autism Hug Machine Portable Seat. J. Intellect. Disabil. - Diagnosis Treat. 2021, 9, 182–188. https://doi.org/10.6000/2292-2598.2021.09.02.4
  1. For participants in the current study, are there standardizations or baselines? because the number of participants and participant criteria will greatly affect the results. This increasingly needs to be highlighted considering the small number of participants and only one referral unspecific autism research service centre which makes the data from the research not become strong and leads to misinterpretation of the results. Sampling criteria for participants also need to be further detailed. The point highlighted is very important and at this point makes me doubt the results, which makes me more likely to recommend rejection for the current manuscript.
  2. In the Results section (line 502-599), the authors are advised to compare the results they obtain with previous similar/identical studies if it is possible.
  3. In the last paragraph before conclusion section (after line 693), the authors should add of one paragraph about the limitations of the presented study
  4. The conclusion (line 694-710) of the present manuscript is not solid. Further elaboration is needed.
  5. Further research needs to be explained in the conclusion section (line 694-710).
  6. In the whole of the manuscript, the authors sometimes made a paragraph only consisting of one or two sentences that made the explanation not clearly understood. The authors need to extend their explanation to become a more comprehensive paragraph. In one paragraph, it is recommended to consist of at least 3 sentences with 1 sentence as the main sentence and the other sentences as supporting sentences. For example in line 453-455.
  7. I see some errors on English in some areas of the present manuscript. To improve the quality of English used in this manuscript and make sure English language, grammar, punctuation, spelling, and overall style are correct, further proofreading is needed. As an alternative, the authors can use the MDPI English proofreading service for this issue.
  8. Please make sure the authors have used the Brain Science, MDPI format correctly. The authors can download published manuscripts by Brain Science, MDPI, and compare them with the present author's manuscript to ensure typesetting is appropriate.

I am pleased to have been able to review the author's present manuscript. Hopefully, the author can revise the current manuscript as well as possible so that it becomes even better. Good luck for the author's work and effort.

Best regards,

The Reviewer

Reviewer 3 Report

File attached.

Round 2

Reviewer 2 Report

Dear Zhou et al.,

After carefully reading the author's revised manuscript entitled "Attentional Engagement and Disengagement Differences for Circumscribed Interest Objects in Young Chinese Children with Autism Spectrum Condition: An Eye Movement Study" (brainsci-1698532) by Zhou et al., The current manuscript does not provide a significant improvement after major revision with very minimum effort. Also, the authors are failing to address all of the fundamental critical comments regarding their manuscript.

The originality, state of the art, research novelty, and research significance are questioned and can be said to be non-existent because they have been achieved by published articles based on my best knowledge and expertise in this field. And, this is also reinforced by several similar articles that have been published by previous authors, both those included in the reference of the current manuscript and those recently published by the author and not listed in the reference. This is exacerbated by the author's inability to make a constructive explanation regarding the contribution of the current manuscript by systematically explaining the research gap.

In addition, the use of research samples on children with autism spectrum disorders from China was evaluated as not providing a serious scientific contribution because it can be represented by a sample of Asian children who have been widely adopted and published in various related literature. This renders the current publication unsubstantiated, even after I saw the author's explanation and reviewed the revised manuscript.

I also criticize the problem of the small number of samples because it greatly affects the three tests carried out in the current study. Although this has been explained previously by the authors, it does not rule out the limitation of the current manuscript which affects the results that are dangerous to the scientific community. The descriptions of the three experiments contained in the manuscript are also not detailed, so more detailed explanations are needed.

Another substance problem is the CI category which needs to be validated. Eye-tracking does not get a clear description. And the source of the participant is only in one location.

Best regards,

The Reviewer

Round 3

Reviewer 2 Report

Thank you very much for the author's response to the previous review report. I appreciate the author's efforts to rebuttal in this session for every point that I convey. Unfortunately, I do not see any meaningful explanation for the same thing I criticized earlier. With all due respect, my decision remains to recommend this manuscript for rejection.